# Numerical Simulation of Flow Velocity Characteristics during Capsule Hydraulic Transportation in a Horizontal Pipe

**Fei Li, Yongye Li \*, Xihuan Sun and Xiaoni Yang**

College of Water Resources Science and Engineering, Taiyuan University of Technology, Taiyuan 030024, China; lifei@tyut.edu.cn (F.L.); sunxihuan@tyut.edu.cn (X.S.); yangxiaoni0916@126.com (X.Y.)

\* Correspondence: liyongye@tyut.edu.cn; Tel.: +86-139-3423-9832

**Abstract:** Capsule hydraulic transportation is a kind of low-carbon and environmentally friendly pipeline transportation technique. In this study, the flow velocity characteristics in the pipeline when the capsule is transported in a straight pipe section were simulated by adopting the RNG (Renormalization Group) k–ε turbulence model based on Fluent software and experimentally verified. The results showed that the simulated value of flow velocity in the pipeline was basically consistent with the experimental value during transportation of the material by the capsule, and the maximum relative error was no more than 6.7%, proving that it is feasible to use Fluent software to simulate the flow velocity characteristics in the pipeline when the capsule is transported in a straight pipe section. In the process of material transportation, the flow velocity distribution of the cross-section near the upstream and downstream sections of the capsule was basically the same, which increased with the increased length–diameter ratio of the capsule. The axial flow velocity was smaller in the middle of the pipe and larger near the inner wall of the pipe. From the inner wall to the center of the pipe, the radial flow velocity first increased and then decreased. The circumferential flow velocity was distributed in the vicinity of the support body of the capsule. The axial flow velocity of the annular gap section around the capsule first increased and then decreased from the inner wall of the pipe to the outer wall of the capsule. In the process of transporting materials, the influence of the capsule on the flow of its downstream section was greater than that of its upstream section. These results could provide a theoretical basis for optimizing the technical parameters of capsule hydraulic transportation.

**Keywords:** flow velocity characteristics; hydraulic transportation; capsule; pipeline

## 1. Introduction

With the steady improvement of social and economic construction, people's demand for the development of the transportation industry is increasing, but the environmental pollution caused by energy consumption and carbon emissions is also increasing [1]. In recent years, energy consumption and carbon emissions generated by the transportation industry have grown rapidly, which makes it one of the industries with high energy consumption and high emissions [2–4]. The proposal of the five national development concepts also reflects the national attention to green development [5]. Especially with the condition of extensive severe haze weather in recent years in China, it is particularly important to seek ways to solve or reduce the carbon emission problem of traditional transportation [6]. Among them, the development of new modes of transportation, such as pipeline transportation, is one effective way to ease transportation tension and reduce carbon emissions [7,8].

Pipeline hydraulic transportation is a branch of pipeline transportation [9]. At present, pipeline hydraulic transportation modes can be divided into slurry, mold, and capsule according to the

characteristics of materials transported [10]. Compared to the traditional mode of transportation, pipeline hydraulic transportation technology has the advantages of low cost, small footprint, low pollution, and no carbon emissions in the transportation process [11]. At the same time—combined with digital control—information management and automatic control can be realized. At home and abroad, it has been widely used in mineral and slag transport, sediment dredging, and other fields. However, there are still many defects in the slurry and mold pipeline hydraulic transportation modes. For example, due to the direct contact between the material and the carrier liquid, it is required that the transported material should not react with the carrier liquid. The slurry can cause wear to the pipeline during transportation, and settling can occur [12]; in the process of transportation, the stability of the molding material cannot be guaranteed, which results in a collision between the molding material and the pipeline and reduces the service life of the pipeline. In addition, before transporting the materials, a binder is needed to make the materials bond and form, and, after being transported to the destination, the materials need to be degummed and crushed, resulting in increased transportation cost [13]. However, capsule pipeline hydraulic transportation technology, to some extent, can alleviate the above defects of the existing technology. First, because of its special structure, the center of the capsule always coincides with the center of the pipeline in the process of transporting the material. That is, to say, the capsule always moves along the central axis of the pipeline in the transporting process, which increases the stability of the material being transported. Second, the material is packed in the capsule, which ensures that transportation technology is a pure material transporting mold, thus saving the transportation cost. Third, transportation technology does not need to add a drag-reducing agent and a material binder and will not pollute the environment. Therefore, capsule pipeline hydraulic transportation technology is a kind of low-carbon, environmentally friendly green transportation mode.

As a new solid–liquid separation technology, capsule pipeline hydraulic transportation technology is a mode to transport the required materials in a columnar capsule to the destination by pipeline. Liu [14] designed and studied the delivery device of the capsule pipeline hydraulic transportation system and discussed the economy of the technology in the application. After that, many relevant scholars have also analyzed and optimized the power device and capsule fabrication process in the capsule transportation system, further improving the technology [15]. Ulusarslan [16] studied the effects of the size and arrangement of the capsule on its movement speed. Kollár et al. [17] studied the relationship between the movement velocity of the capsule and its geometric parameters and Reynolds number and obtained the optimal diameter of the capsule based on the minimum energy consumption. Agarwal et al. [18–20] analyzed the effects of the size of the capsule structure on the movement of the capsule and concluded that the ratio of capsule diameter to pipe diameter had the most significant effect on the movement speed of the capsule. Li et al. [21–23] studied the movement characteristics of the capsule under different length–diameter ratios, flow discharges, and transporting load conditions, established a movement model of the capsule under various influencing factors, and analyzed the energy consumption of the capsule in the process of transporting materials. Ulusarslan and Teke [24,25], as well as Ulusarslan [16], studied the pressure drop of flow during the movement of a low-density spherical capsule and analyzed the relationship between the pressure drop of flow and the density of the capsule. Ginevskii et al. [26] analyzed the relationship between the flow pattern and the movement velocity of the capsule and concluded that when the flow pattern was turbulent, the velocity of the capsule was greater than the average velocity of the carrier. Ooms et al. [27] studied the buoyancy effect's characteristics of horizontal core-annular flow based on the finite volume method and concluded the different causes of the annular layer core levitation at high Reynolds number and low Reynolds number and how the buoyancy force on the core was counterbalanced. Sotgia et al. [28] studied the characteristics of water continuous oil-water flows, analyzed the influence on the two-phase flow features of the inlet mixer, pipe material, and its history, and concluded a criterion for the location of boundaries between the annular and stratified flow. Lee et al. [29] studied two-phase core-annular flow in a horizontal pipe based on the level set method and analyzed the core fluid movement characteristic and the oil-water interface. Jing et al. [30] and Zhang et al. [31] studied the hydraulic characteristics

of annular gap flow generated by the movement of the capsule under different Reynolds numbers, analyzed the relationship between the average velocity of annular gap flow, the velocity of the capsule, and the average velocity of flow in the pipeline and Reynolds number, and determined the evaluation criteria for the optimal Reynolds number. Based on the 6DOF (six-degree-of-freedom) coupling model, Zhang et al. [32,33] studied the pressure characteristics of annular gap flow when the capsule was still in the pipeline and concluded that the pressure of annular gap flow first decreased, then increased, and then decreased from the inlet to the outlet. However, these studies do not provide sufficient information regarding the flow velocity characteristics in the pipe during the transport of the capsule in order to industrialize the technique of capsule hydraulic transportation. Therefore, in this study, the flow velocity characteristics during capsule hydraulic transportation in a straight pipe section were investigated, aiming to provide a theoretical basis for optimizing capsule hydraulic transportation technical parameters.

## 2. Mathematical Model

### 2.1. Fluid Domain Governing Equations

In the process of transporting materials, considering the complexity of flow in the pipeline during the movement of the capsule, the following assumptions were made in the numerical calculation process: (1) ignore the change of density of the fluid medium and regard the physical parameters of the fluid as constants and (2) ignore the heat exchange between the fluid and the inner wall of the pipe. The continuity equation and Reynolds mean momentum equation were used for calculation [34], which could be expressed as

$$\frac{\partial \rho}{\partial t} + \frac{\partial}{\partial x_i}(\rho u_i) = 0 \tag{1}$$

$$\frac{\partial(\rho u_i)}{\partial t} + \frac{\partial}{\partial x_j}\left(\rho u_i u_j\right) = -\frac{\partial p}{\partial x_i} + \frac{\partial}{\partial x_j}\left(\mu \frac{\partial u_i}{\partial x_j} - \rho \overline{u_i' u_j'}\right) + S_b \tag{2}$$

where $t$ is time (s); $\rho$ is the density of the fluid in the pipe (kg·m$^{-3}$); $u_i$ and $u_j$ are the mean values of velocity in the $i$ and $j$ directions, respectively (m·s$^{-1}$); $u_i'$ and $u_j'$ are the pulsation values of the velocity components in the $i$ and $j$ directions, respectively (m·s$^{-1}$); $i$ and $j$ are 1, 2, and 3, respectively; $p$ is the mean value of pressure (Pa); $\mu$ is the dynamic viscosity of the fluid (Pa·s); $x_i$ and $x_j$ are coordinate components (m); $S_b$ is the generalized source term of the momentum equation; $\rho \overline{u_i' u_j'}$ is Reynolds stress.

The relationship between Reynolds stress and the average velocity gradient was proposed by the vortex viscosity hypothesis, which could be expressed as

$$\rho \overline{u_i' u_j'} = \left[\mu_i\left(\frac{\partial u_i}{\partial x_j} + \frac{\partial u_j}{\partial x_i}\right)\right] - \frac{2}{3}\rho_w k \delta_{ij} \tag{3}$$

where $\mu_i$ is turbulent viscosity (Pa·s); $\varepsilon$ is the turbulence dissipation rate (m$^2$·s$^{-3}$); $k$ is turbulent kinetic energy (m$^2$·s$^{-2}$); $\delta_{ij}$ is the Kronecker delta ($\delta_{ij} = 1$ when $i = j$; $\delta_{ij} = 0$ when $i \neq j$).

In the process of transporting materials, the movement of the capsule could be divided into three stages: the acceleration stage, stable operation stage, and the deceleration stage. In this paper, the flow velocity characteristics during the stable movement of the capsule in a horizontal pipe were studied.

In this study, the flow discharge in the pipe was 40 m$^3$/h, and the diameter of the transporting pipe was 100 mm. Through calculation, it was found that the Reynolds number of the flow in the pipe was greater than 2320, indicating that the flow was in a turbulent state. At the same time, in the process of movement of the capsule through the action of flow in the pipeline, due to the influence of special structures, such as the support, flow around the capsule would occur, and the flow field was relatively complex and was not constant. Therefore, the RNG (Renormalization Group) k–$\varepsilon$ turbulence model in Fluent software was adopted in this paper to simulate the flow velocity characteristics during

the transport of the capsule. The distances between the capsules with each length–diameter ratio and the pipeline inlet were the same in the simulation process.

## 2.2. Solid Domain Governing Equations

The capsule studied in this paper was a rigid structure. Therefore, when the capsule was subjected to flow in the pipeline during movement, its shape, size, and relative position of internal points did not change. The structural coupling characteristics of the capsule were mainly reflected in the instantaneous displacement and velocity of the capsule. The displacement of the capsule under the action of the fluid produced a translational motion along the pipe. The instantaneous velocity of the capsule changed with time under the action of fluid pulsation pressure. The rigid motion equation could solve the instantaneous migration velocity and instantaneous displacement of the capsule at any time. The rigid motion equation could be expressed as

$$
\begin{aligned}
F &= ma \\
M &= I\alpha + \omega \times (I \cdot \omega)
\end{aligned}
\tag{4}
$$

where $m$ is the mass of the capsule (kg); $F$ is the resultant force on the capsule (N); $a$ is the instantaneous acceleration of the capsule (m·s$^{-2}$); $M$ is the instantaneous moment of the capsule (N·m); $I$ is the matrix of instantaneous rotational inertia of the capsule; $\omega$ is the instantaneous angular velocity of the capsule (rad·s$^{-1}$); $\alpha$ is the instantaneous angular acceleration of the capsule (rad·s$^{-2}$). Newmark's predictor–corrector implicit time integration algorithm was used to solve the rigid motion equation. In the coupling interface between the fluid domain and solid domain, the coupling variables between instantaneous displacement and instantaneous stress of the two domains should be equal or conserved. The governing equation for the coupling variables could be expressed as

$$
n_s \tau_f = n_s \tau_b, \; r_f = r_b
\tag{5}
$$

where $\tau$ and $r$ are instantaneous stress and instantaneous displacement at the coupling interface, respectively; $n_s$ is the normal vector; $f$ is the fluid domain; $b$ is the solid domain.

## 2.3. Dynamic Mesh Model

The dynamic mesh model was mainly used to adjust the nodes of the computational mesh. In Fluent software, there are three dynamic mesh algorithms: dynamic layered algorithm, elastic smooth algorithm, and local reconstruction algorithm. In this paper, two of these models, elastic smooth and local reconstruction, were used to complete the research of dynamic mesh numerical simulation.

The law of conservation of mass should be followed in the calculation of a dynamic mesh model. The conservation equation of general scalar $\phi$ on any control volume $V$ with boundary movement could be expressed as follows:

$$
\frac{d}{dt}\int_V \rho\phi dV + \frac{d}{dt}\int_{\partial V} \rho\phi\left(\vec{u} - \vec{u}_g\right)d\vec{A} = \frac{d}{dt}\int_{\partial V} \Gamma\nabla\phi \cdot d\vec{A} + \frac{d}{dt}\int_V S_\phi dV
\tag{6}
$$

where $V(t)$ is the control volume, whose scope and shape change with time in space; $\partial V(t)$ is the motion boundary of the control volume; $\vec{u}_g$ is the motion velocity of the moving grid; $\rho$ is the density of the fluid; $\vec{u}$ is the velocity vector of the fluid; $\Gamma$ is the dissipation coefficient of the equation; $S_\phi$ is the source term of the scalar $\phi$.

In Equation (6), the time derivative term could be obtained by using the first-order backward difference formula:

$$
\frac{d}{dt}\int_V \rho\phi dV = \frac{(\rho\phi V)^{n+1} - (\rho\phi V)^n}{\Delta t}
\tag{7}
$$

where the superscripts $n$ and $n + 1$ represent, respectively, the current and next time layers.

The $(n + 1)$ time step control volume $V^{n+1}$ could be obtained from Equation (8):

$$V^{n+1} = V^n + \frac{dV}{dt} \Delta t \tag{8}$$

$$\text{where } \frac{dV}{dt} = \int_{\partial V} \vec{u}_g \cdot d\vec{A} = \sum_j^n {}^j \vec{u}_{g,j} \cdot \vec{A}_j \tag{9}$$

where $\vec{A}_j$ is the $j$th area vector, and $\vec{u}_{g,j} \cdot \vec{A}_j$ is the dot product of each control surface, which could be expressed as

$$\vec{u}_{g,j} \cdot \vec{A}_j = \frac{\delta V_j}{\Delta t} \tag{10}$$

where $\delta V_j$ is the volume of surface $j$ on the control body moving in time step $\Delta t$.

### 2.4. Model Establishment

In this paper, the velocity characteristics of the flow were simulated in a straight pipe when the capsule was transported in the pipe. In order to verify the results, the simulation conditions were consistent with the test conditions. The diameter of the pipeline was 100 mm, the length–diameter ration of the two investigated capsules were $L/D = 2$ and $L/D = 1.67$, the transporting load was $m = 1150$ g, and the flow discharge was 40 m³/h. The mathematical model of the transport of the capsule in the straight pipe in the Cartesian coordinate system is shown in Figure 1, and the basic coordinates of the mathematical model are shown in Table 1. A physical diagram of the capsule is shown in Figure 2.

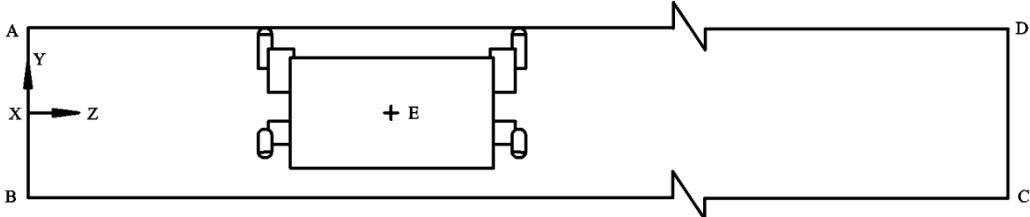

**Figure 1.** Mathematical model of capsule transport in a straight pipe.

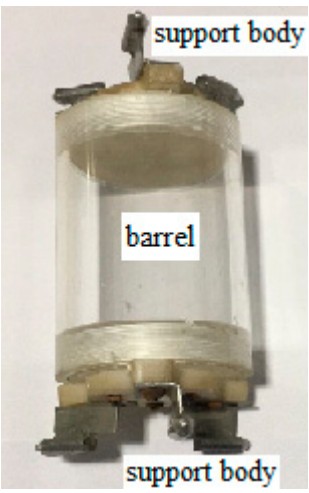

**Figure 2.** Physical diagram of the capsule.

**Table 1.** Basic coordinates of the mathematical model (unit: m).

| | | | | | |
|---|---|---|---|---|---|
| A | (0, 0.05, 0) | B | (0, −0.05, 0) | C | (0, −0.05, 11.0) |
| D | (0, 0.05, 11.0) | E | (0, 0, 0.3) | | |

Figure 1 shows the locations of the internal and external boundaries of the calculation domain, where AB represents the boundary of the inlet section of the model, CD represents the boundary of the outlet section of the model, and AD and BC represent the location of the internal wall of the pipeline.

## 2.5. Mesh Division

Through observation of the calculation domain and the structure of the capsule, it could be seen that the structure of the capsule was composed of a large cylinder connected with six small cylinders. The unpassed part of the capsule in the computational domain was a standard cylinder with a single structure, so a combined grid was adopted. A regular tetrahedral unstructured mesh was used to divide the capsule, and an O-shaped unstructured mesh was used to divide the drainage area in the pipeline without the capsule. Finally, the two meshes were assembled to form a complete set of meshes. The specific mesh division is shown in Figure 3. In this paper, Gambit preprocessing software was used to encrypt the volume grid of the computational domain mathematical model. It was found that when the volume mesh size was 0.003 m, the velocity in three directions of Cartesian coordinates could reach the range of accuracy required by the calculation, and the volume mesh number of the calculation model was the least. Therefore, the volume mesh encryption size of the calculation domain mathematical model adopted in this paper was 0.003 m. The encryption inspection of volume mesh is shown in Table 2.

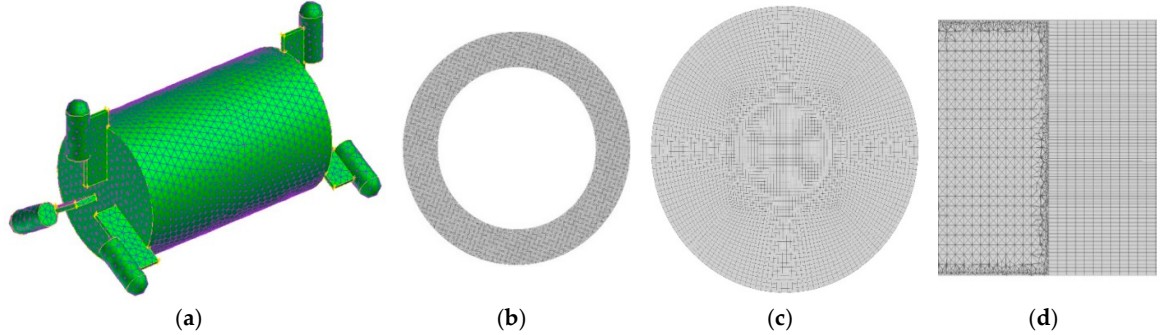

(**a**)          (**b**)          (**c**)          (**d**)

**Figure 3.** Schematic diagram of mesh division: (**a**) capsule surface mesh; (**b**) annular surface mesh; (**c**) O-shaped structured mesh; (**d**) interface merging mesh.

**Table 2.** The encryption inspection of volume mesh.

| Volume Mesh Size | Grid Cell | Velocity at the Coordinate (0, 0, 1) | | |
|---|---|---|---|---|
| | | $V_x$ (m/s) | $V_y$ (m/s) | $V_z$ (m/s) |
| 0.008 m | 1514464 | 0.000161284 | −0.0036124 | 1.49384 |
| 0.007 m | 1942178 | 0.000081247 | −0.0019522 | 1.47659 |
| 0.006 m | 2514769 | 0.000042387 | −0.0013584 | 1.46554 |
| 0.005 m | 3238411 | 0.000039584 | −0.0006253 | 1.45831 |
| 0.004 m | 3745843 | 0.000024927 | −0.0004855 | 1.45623 |
| 0.003 m | 4319562 | 0.000024315 | −0.0002516 | 1.45447 |
| 0.002 m | 5684124 | 0.000024113 | −0.0002436 | 1.45433 |
| 0.001 m | 9268415 | 0.000024132 | −0.0002411 | 1.45431 |

*2.6. Boundary Conditions*

The boundary conditions of the mathematical model selected in this paper mainly included inlet boundary, outlet boundary, wall boundary, mesh interface, and dynamic boundary.

Inlet boundary: The flow velocity inlet boundary was adopted. Since the flow discharge selected in the simulation was 40 m³/h, the initial flow average velocity at the inlet was 1.415 m/s.

Outlet boundary: The pressure outlet boundary was adopted. Since the pressure at the outlet section of the pipeline was atmospheric pressure and the flow was free flow, the relative pressure at the outlet section was set to 0.

Wall boundary: The wall boundary was adopted for the inner wall of the pipeline and the surface of the capsule.

Mesh interface: The mesh interface was set to the interior boundary so that the fluid could pass through.

Moving boundary: The wall of the capsule was a dynamic boundary condition, and the sliding free boundary condition was adopted for the moving boundary wall. Capsule movement belonged to the category of rigid movement. The velocity of the capsule was measured by experiments as the moving velocity of the wall boundary of the movable capsule.

## 3. Model Validation and Result Analysis

*3.1. Model Validation*

In order to verify the reliability of the numerical simulation results, an experimental study was conducted of the flow velocity characteristics in the pipe when the capsule was transported in the straight pipe section. The test device [35–37] was mainly composed of the centrifugal pump of the water supply device, the test pipeline, the capsule delivery device and receiving device, the turbine flow meter for measuring the flow discharge, the infrared sensor for measuring the capsule velocity, and the PDA (particle image velocimetry) for measuring the flow velocity in the pipeline. A schematic diagram of the test device is shown in Figure 4. During the test, water was pumped out of the underground reservoir by a centrifugal pump and flowed into the plexiglass pipe through the steel pipe. The capsule was placed into the test pipe from the delivery device and secured by the brake device. The flow discharge was adjusted through the gate valve to the flow discharge required by the test. After the flow was stable, the braking device was removed, the capsule was released, and the movement velocity of the capsule and flow velocity in the pipeline during the movement of the capsule in the test section were measured. Finally, the capsule entered the receiving device from the pipe outlet, and the water flowed into the underground reservoir through the outlet pool, forming a circulation system.

The selected sections were located 0.15 m, 0.3 m, and 0.45 m from the downstream section of the capsule and 0.15 m from the upstream section. The length–diameter ratios of the capsule were 2 and 1.67, and the flow discharge was 40 m³/h. The simulation and experimental results of the axial flow velocity in the pipeline during the material transport process were compared and analyzed. The results are shown in Figure 5.

As could be seen from Figure 3, the simulation results of the axial flow velocity in the pipeline were basically consistent with the test results during the process of transporting the material in the capsule, and the maximum relative error was no more than 6.7%, indicating that the flow velocity characteristics in the pipeline simulated by Fluent were feasible when the capsule was transported in the straight pipeline.

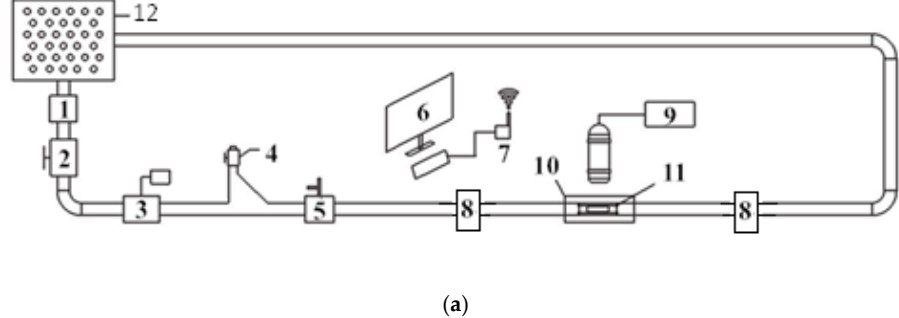

(**a**)

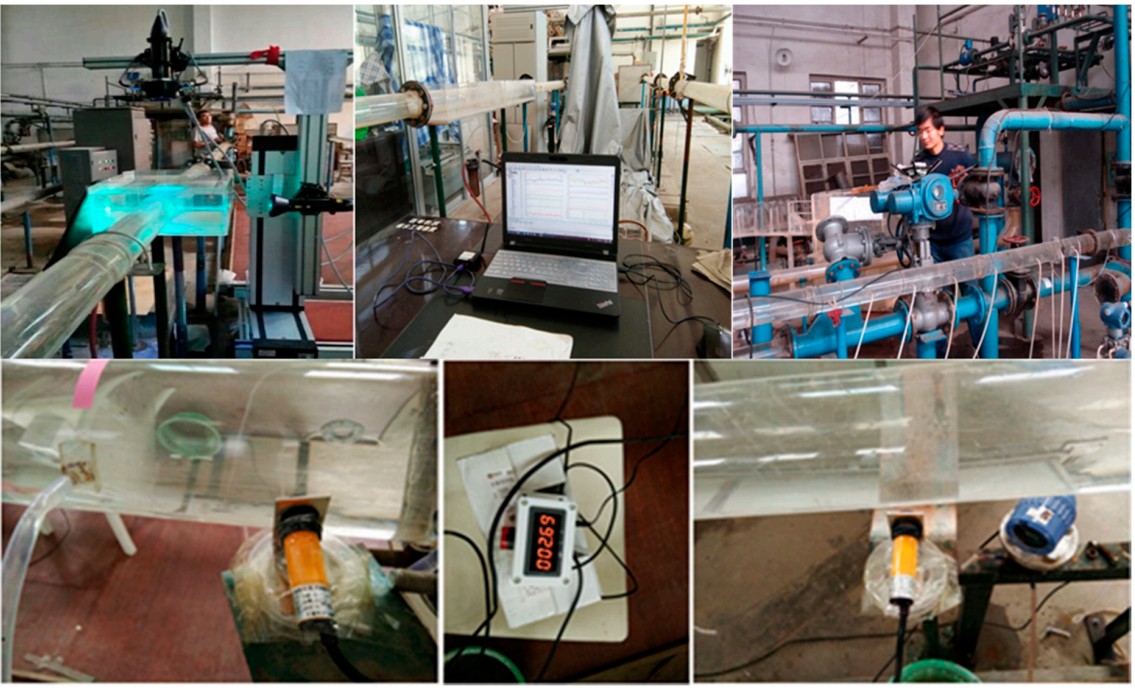

(**b**)

**Figure 4.** Experimental system device: (**a**) schematic diagram: 1. centrifugal pump; 2. regulating valve; 3. turbine flow meter; 4. capsule delivery device; 5. brake device; 6. computer; 7. flow velocity receiving device; 8. infrared sensor; 9. particle image velocimetry (PDA); 10. rectangular water jacket; 11. capsule; 12. water supply device and capsule receiving device; (**b**) physical diagram.

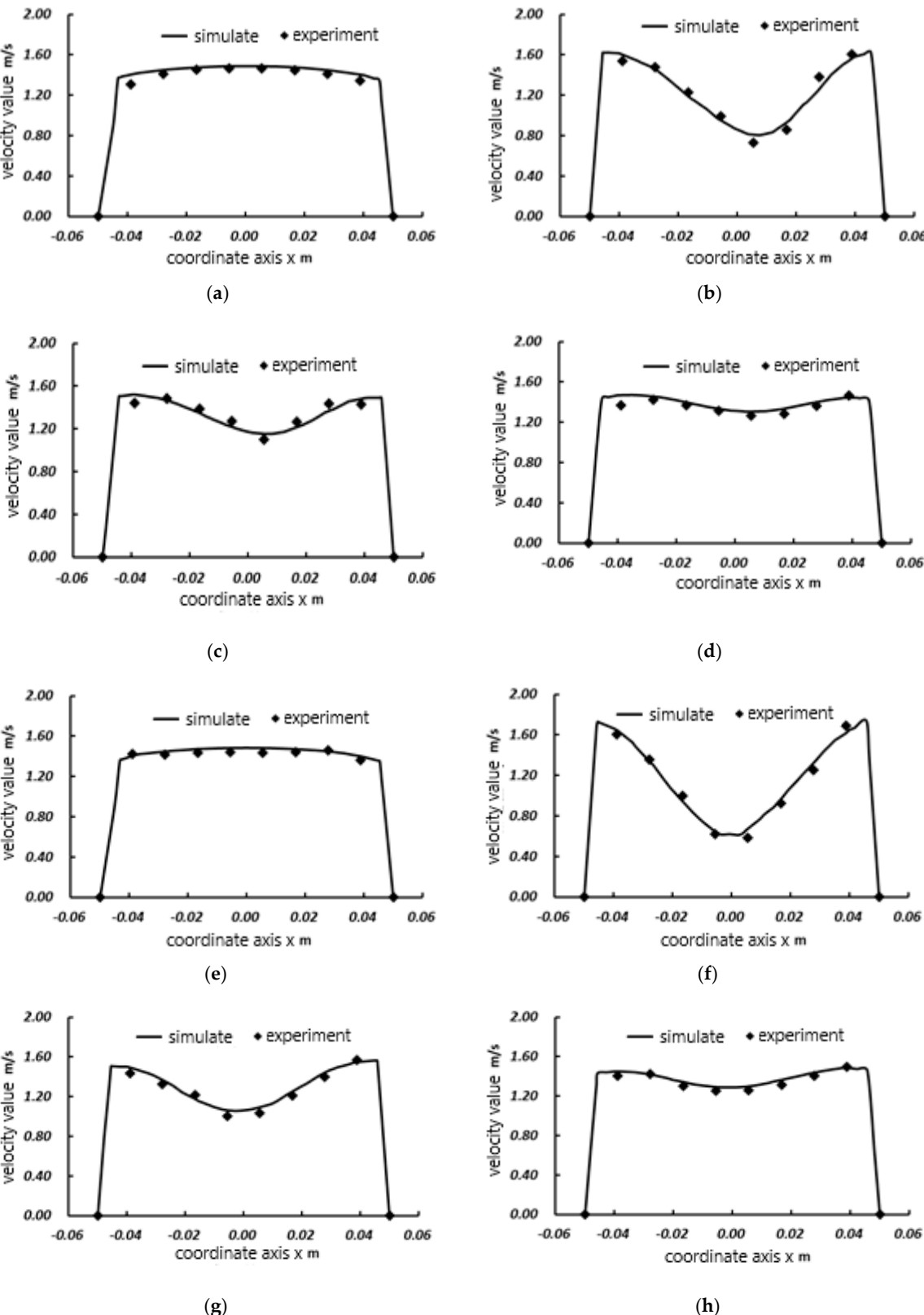

**Figure 5.** Experimental and simulation results of axial flow velocity during the process of transporting material in the capsule: (**a**) length–diameter ratio (*L/D*) = 2, the upstream section of the capsule $Z_u$ = 0.15 m; (**b**) *L/D* = 2, the downstream section of the capsule $Z_d$ = 0.15 m; (**c**) *L/D* = 2, $Z_d$ = 0.30 m; (**d**) *L/D* = 2, $Z_d$ = 0.45 m; (**e**) *L/D* = 1.67, $Z_u$ = 0.15 m; (**f**) *L/D* = 1.67, $Z_d$ = 0.15 m; (**g**) *L/D* = 1.67, $Z_d$ = 0.30 m; (**h**) *L/D* = 1.67, $Z_d$ = 0.45 m.

### 3.2. Result Analysis

In the process of transporting materials, the existence of the capsule would lead to the phenomenon of circumfluence, and the flow between the capsule and the wall was concentric annular gap flow, which all changed the velocity distribution of the original flow in the pipeline. The flow velocity distribution of the upstream and downstream sections of the capsule and the annular gap sections between the capsule and the pipe and the influence range of the capsule on the flow were studied based on the fundamentals of two-fluid dynamics and the theory of hydraulics [38,39].

3.2.1. Flow Velocity Characteristics of the Downstream Section of the Capsule

Taking $Z = 0.075$ m and $Z = 0.225$ m from the downstream sections of the capsule as examples, the velocity distribution of the downstream sections during material transportation was analyzed.

Figure 6 shows the distributions of the axial flow velocity of the downstream sections of the capsules, as follows:

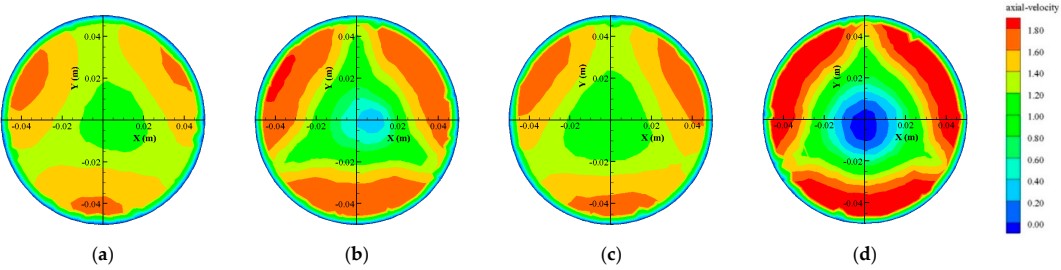

**Figure 6.** Axial flow velocity distributions of the downstream sections of capsules. (**a**) $Z_d = 0.225$ m, $L/D = 2$; (**b**) $Z_d = 0.075$ m, $L/D = 2$; (**c**) $Z_d = 0.225$ m, $L/D = 1.67$; (**d**) $Z_d = 0.075$ m, $L/D = 1.67$.

1.  For capsules with the same length–diameter ratio, the axial flow velocities of the downstream sections were smaller in the middle area of the pipe and larger near the inner wall of the pipe. The farther away from the downstream section of the capsule, the smaller the area of low-velocity in the middle and high-velocity in the inner wall. This was mainly because farther from the downstream section of the capsule, the velocity of the local jet formed by the annular gap flow between the capsule and the pipeline decreased in the downstream pipeline, and the flow velocity in the middle low-velocity area gradually became the flow velocity in the downstream pipeline, so the average flow velocity in the downstream section of the capsule gradually approached the average flow velocity in the pipeline. The farther from the downstream section of the capsule, the smaller the overall difference of velocity.

2.  For the same downstream section of the capsule, the axial flow velocity of the same location increased with the increased length–diameter ratio of the capsule. Moreover, the area of low-flow velocity in the downstream section of the capsule was greatly affected by the length–diameter ratio. As for the two capsules in this paper, the larger the capsule's length–diameter ratio, the smaller the influence range of the low-flow velocity area. This was mainly related to the generation of the low-velocity region and the local jet formed by annular gap flow. The greater the length–diameter ratio of the capsule, the greater the local jet area formed between the capsule and the pipe, and the smaller the cavity area formed downstream of the capsule, resulting in a decreased area of downstream flow randomly filling the mainstream cavity area.

3.  The flow velocity distribution in the downstream section of the capsule was greatly affected by the support body of the capsule section. Since the front and rear faces of the capsule were installed with cylindrical supports at equal interval angles of 120°, the high-velocity area was divided into three central symmetrical areas.

Figure 7 shows the distributions of the radial flow velocity of the downstream sections of the capsules, as follows:

1. For the same flow discharge condition, the variation of radial flow velocity in the downstream section of the capsule with different length–diameter ratios was basically the same, and the radial velocity from the inner wall of the pipe to the center of the pipe showed a changing trend of first increasing and then decreasing. At the same time, the annular gap flow caused a local cavity in the center of the pipeline, and the radial velocity was small, while there was no deformation near the wall boundary, and the streamline did not change greatly. For the annular region in the middle of the pipe, the sudden expansion of the streamline caused the flow to jet into the center of the pipe, which resulted in the larger radial velocity of the annular region in the middle compared with other regions. Moreover, the radial flow velocity of the annular region in the middle was basically negative, indicating that the flow direction along the radial direction was from the inner wall to the center of the pipe.

2. For the same downstream section of the capsule, the radial flow velocity at the same location increased with an increased length–diameter ratio of the capsule. This was mainly because the radial flow velocity of the downstream section was related to the convergence of the annular gap flow to the downstream pipeline. The larger the length–diameter ratio, the larger the volume of the downstream cavity formed between the capsule and the pipe, the greater the convergence degree of the annular gap flow, and the larger the radial flow velocity.

3. Farther from the downstream section of the capsule, the kinetic energy generated by the annular gap flow formed between the capsule and the pipeline decreased gradually, and the interaction between the annular gap flow and the flow in the downstream pipeline resulted in the inconsistency of the radial flow velocity distribution in the downstream section.

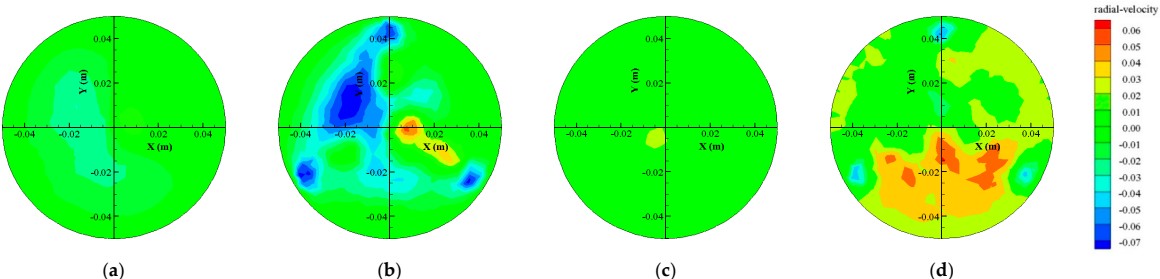

**Figure 7.** Radial flow velocity distributions of the downstream sections of capsules. (**a**) $Z$ = 0.225 m, $L/D$ = 2; (**b**) $Z$ = 0.075 m, $L/D$ = 2; (**c**) $Z$ = 0.225 m, $L/D$ = 1.67; (**d**) $Z$ = 0.075 m, $L/D$ = 1.67.

Figure 8 shows the distributions of the circumferential flow velocity of the downstream sections of the capsules, as follows:

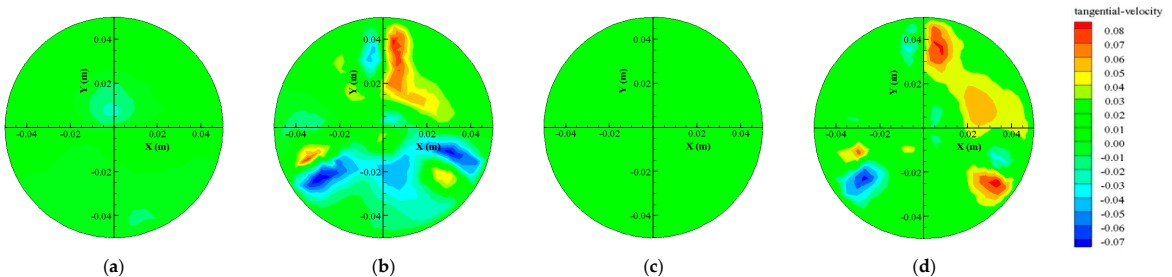

**Figure 8.** Circumferential velocity distributions of the downstream sections of capsules. (**a**) $Z$ = 0.225 m, $L/D$ = 2; (**b**) $Z$ = 0.075 m, $L/D$ = 2; (**c**) $Z$ = 0.225 m, $L/D$ = 1.67; (**d**) $Z$ = 0.075 m, $L/D$ = 1.67.

1. For the same flow conditions, the changing trend of circumferential flow velocity in the downstream section of the capsule with different length–diameter ratios was basically the same, which was mainly distributed around the support body of the capsule. Moreover, at the

left and right positions of the support body, the circumferential flow velocity presented the symmetrical distribution law in the opposite direction. At the same time, the farther from the downstream section of the capsule, the smaller the tangential velocity of the downstream section.

2. For the same downstream section of the capsule, the circumferential flow velocity at the same location increased with an increased length–diameter ratio of the capsule. The main reason was that the circumferential velocity of the downstream section was related to the influence degree of the annular gap flow around the support body of the capsule. The larger the length–diameter ratio, the greater the influence of the annular gap flow around the support body of the capsule, and the greater the circumferential flow velocity.

### 3.2.2. Flow Velocity Characteristics of the Upstream Section of the Capsule

Taking $Z = 0.105$ m and $Z = 0.075$ m from the upstream sections of the capsule as examples, the velocity distributions of the upstream sections during material transportation were analyzed.

Figure 9 shows the distributions of the axial flow velocity of the upstream sections of the capsules, as follows:

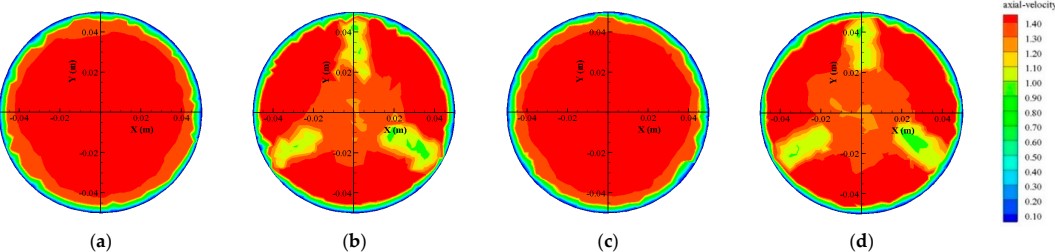

(a)          (b)          (c)          (d)

**Figure 9.** Axial flow velocity distributions of the upstream sections of capsules. (**a**) $Z = 0.105$ m, $L/D = 2$; (**b**) $Z = 0.075$ m, $L/D = 2$; (**c**) $Z = 0.105$ m, $L/D = 1.67$; (**d**) $Z = 0.075$ m, $L/D = 1.67$.

1. For the same section upstream of the capsule, the smaller the length–diameter ratio of the capsule, the smaller the axial flow velocity at the same location. The main reason was that the axial flow velocity in the upstream section was related to the length–diameter ratio. The smaller the length–diameter ratio of the capsule, the larger the action area blocking the axial movement, and the more significant the effect of decreasing the axial velocity. The influence of the capsule on the axial flow velocity in the upstream section was small and only located in the area near the wall of the upstream section.

2. The flow velocity distribution in the upstream section of the capsule was greatly affected by the support body of the capsule section. Since the front and rear faces of the capsule were installed with cylindrical supports at equal interval angles of 120°, the axial flow velocity was divided into three central symmetrical areas, and the low-flow velocity region was located at the support body position of the capsule. The main reason was that when water flowed around the cylindrical support body, a local circumfluence phenomenon occurred, which reduced the kinetic energy of the flow and increased the pressure energy so that a low-velocity zone appeared at the position of the support body of the capsule. However, the area of the low-flow velocity region caused by the support body of the capsule was basically the same.

3. Under the same flow discharge condition, the larger the length–diameter ratio of the capsule, the smaller the influence distance of the capsule on the upstream flow. The main reason was the existence of the capsule, which made the flow in the pipeline fluctuate greatly when passing through the capsule. Moreover, the larger the length–diameter ratio, the smaller the influence range of the wave on the water flow, and the smaller the influence distance on the flow in the upstream section of the capsule.

Figure 10 shows the distributions of the radial flow velocity of the upstream sections of the capsules, as follows:

1.  For the same flow discharge conditions, the variation trend of radial flow velocity in the upstream section of the capsule with different length–diameter ratios was basically the same. On the whole, the radial flow velocity from the inner wall to the center of the pipe first increased and then decreased. The radial flow velocity in the area of the center and the inner wall of the pipe was basically zero.
2.  The cylindrical support body was installed at the front and rear faces of the capsule at equal interval angles of 120°. The radial flow velocity distribution area was divided into three regions with basically equal areas for the influence of the cylindrical support body, and the smaller the length–diameter ratio of the capsule, the larger the distribution area of radial velocity.
3.  Under the same flow conditions, the larger the length–diameter ratio of the capsule, the shorter the influence range of the radial flow velocity of the upstream section of the capsule. At the same time, regardless of the transporting condition of the capsule, the farther from the upstream section of the capsule, the smaller the radial velocity of the upstream section of the capsule.
4.  Under the same transportation conditions, the area of the radial low-flow velocity area in the center of the pipeline increased with the increased length–diameter ratio of the capsule. The main reason was that the flow distance of the fluid group in the upstream section of the capsule became longer for the length–diameter ratio and the interaction between the front and back fluid groups became more obvious, thus the obvious radial movement was formed.
5.  For the same section upstream of the capsule, the radial flow velocity at the same location was positively correlated with the length–diameter ratio of the capsule; that is, the radial flow velocity decreased with the decreased length–diameter ratio.

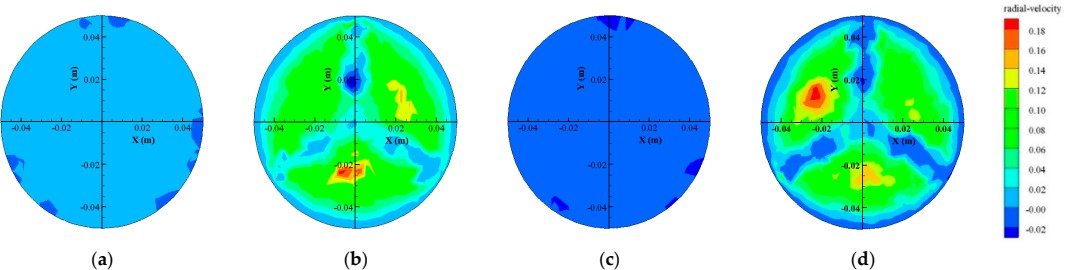

**Figure 10.** Radial flow velocity distributions of the upstream sections of capsules. (**a**) $Z$ = 0.105 m, $L/D$ = 2; (**b**) $Z$ = 0.075 m, $L/D$ = 2; (**c**) $Z$ = 0.105 m, $L/D$ = 1.67; (**d**) $Z$ = 0.075 m, $L/D$ = 1.67.

Figure 11 shows the distributions of circumferential flow velocity of the upstream sections of the capsules, as follows:

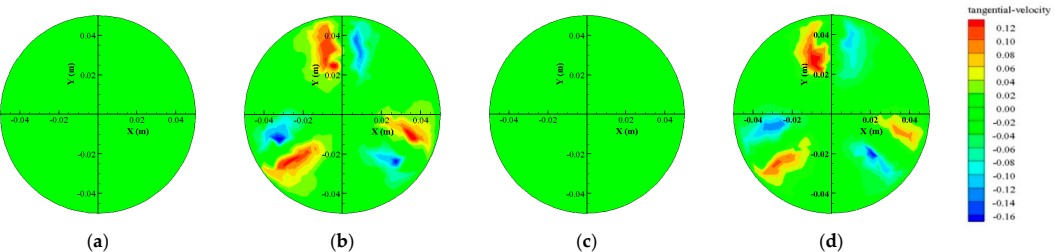

**Figure 11.** Circumferential flow velocity distributions of the upstream sections of capsules. (**a**) $Z$ = 0.105 m, $L/D$ = 2; (**b**) $Z$ = 0.075 m, $L/D$ = 2; (**c**) $Z$ = 0.105 m, $L/D$ = 1.67; (**d**) $Z$ = 0.075 m, $L/D$ = 1.67.

1. The changing trend of circumferential velocity in the upstream section of the capsule with different length–diameter ratios was basically the same as that in the downstream section, which was mainly distributed around the support body of the capsule, showing a symmetrical distribution. Moreover, the circumferential flow velocity around the support body decreased with the decreased length–diameter ratio. At the same time, the farther from the upstream section of the capsule, the smaller the circumferential velocity of the upstream section of the capsule.

2. For the same upstream section of the capsule, the circumferential flow velocity of the same location increased with the increased length–diameter ratio of the capsule. The main reason was that the circumferential flow velocity distribution of the upstream section was related to the flow around the capsule and the support body, and its value was determined by the flow velocity in the pipeline and the movement velocity of the capsule. As the length–diameter ratio decreased, the movement velocity of the capsule increased, and the difference between the velocity of the capsule and the flow velocity in the pipeline became smaller and smaller. Therefore, the circumferential flow velocity at the same position of the upstream section of the capsule decreased gradually with the decreased length–diameter ratio.

### 3.2.3. Axial Flow Velocity Characteristics of Annular Gap Section between Capsule and Pipe

When the capsule was running in the pipeline, the annular gap flow of a certain length was formed along the length direction of the capsule. Taking the capsule with $L/D = 1.67$ and $L/D = 2$ as an example, the axial flow velocity distribution of the annular gap section of the capsule in the process of transporting materials was analyzed when the flow discharge Q = 40 m³/h. The front, middle, and rear sections of the capsule were selected for the study (Figure 12).

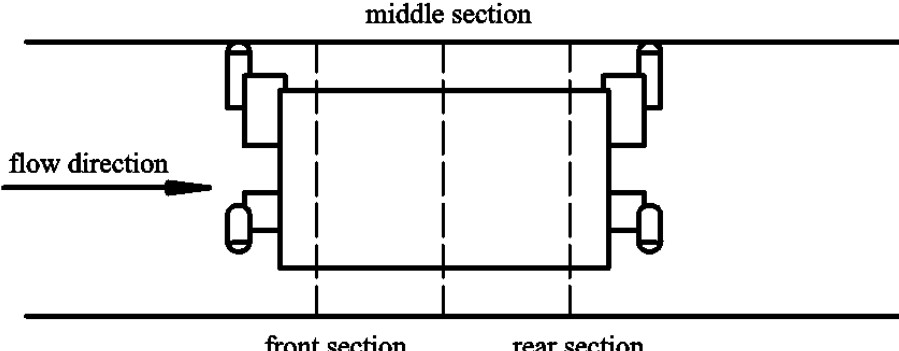

**Figure 12.** Schematic diagram of section division of capsule.

Figure 13 shows the axial flow velocity distribution of the front, middle, and rear annular gap sections of the capsule, as follows:

1. Under the same transportation conditions, the axial flow velocity in the front, middle, and rear annular gap sections of the capsule first increased and then decreased from the inner wall of the pipe to the outer wall of the capsule. However, for the same section, the axial velocity of the annular gap flow in the rear section of the capsule decreased faster, while that in the front section decreased slower.

2. The thickness of the boundary layer formed by the dynamic boundary was obviously smaller than that of the static boundary layer during the material transport of the capsule. The main reason was that the inner wall of the pipeline belonged to the static boundary. Under the action of the viscous force of flow, a stable boundary layer thickness in the inner wall of the pipeline could be formed. However, for the moving capsule, its wall was the dynamic boundary. Relative motion occurred between the fluids at different velocity layers in the moving boundary region, and the viscous force of flow could not act stably on the fluid groups in each layer, resulting in the relatively small dynamic boundary layer thickness.

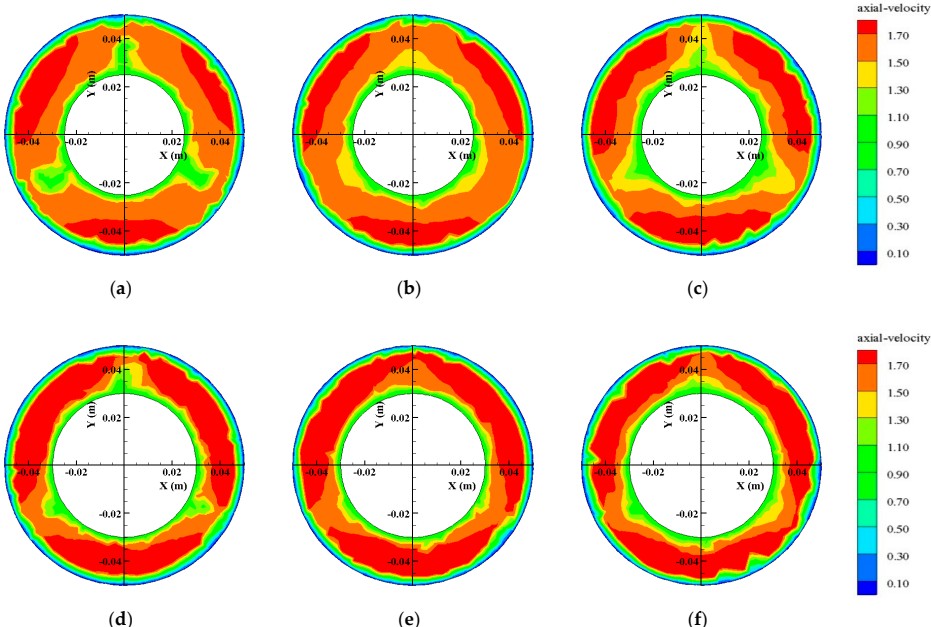

**Figure 13.** Axial flow velocity distribution of annular gap sections of the capsule: (**a**) front section, *L/D* = 2; (**b**) middle section, *L/D* = 2; (**c**) rear section, *L/D* = 2; (**d**) front section, *L/D* = 1.67; (**e**) middle section, *L/D* = 1.67; (**f**) rear section, *L/D* = 1.67.

### 3.2.4. Distribution of Flow Field and its Influence Range during Movement of the Capsule

Figure 14 shows the flow velocity distribution in the pipeline during the material transportation of the capsule, as follows:

1. In the process of transporting materials, the capsule had a small influence on the upstream flow field but a large influence on the downstream flow field. The main reason was that the flow velocity in the pipeline was higher than that of the capsule, which caused local flow around the inside of the fluid. The annular gap flow caused cavities of different sizes in the downstream flow field area of the capsule. The flow in the downstream area of the capsule would randomly fill the region. Under the interaction between the capsule and the flow in the downstream pipeline, the flow velocity of the cavity region was low. However, the flow velocity in the downstream region of the capsule would eventually return to the state of flow velocity when the capsule was not released. Therefore, in the process of conveying materials, the flow field in the downstream region was greatly affected by the capsule.

2. In the process of conveying materials, there was a local axial low-velocity region in the downstream section of the capsule, and the area of this region increased with the increased length–diameter ratio of the capsule.

3. In the process of conveying materials, the radial velocity of the upstream and downstream section near-wall region was larger, and there were two larger areas of radial velocity distribution in the downstream section near-wall region; the radial velocity of the two regions was the same, and the direction was opposite.

4. In the process of transporting materials, the circumferential flow velocity in the pipeline was mainly distributed around the support body in the front and rear sections of the capsule. Moreover, the circumferential flow velocity of the front section of the capsule propagated continuously downstream.

5. In the process of transporting materials, the axial flow velocity at the center of the pipe first decreased and then increased. In the front section of the capsule, the axial flow velocity reached the minimum value. The distribution trends of the circumferential and radial velocity of the flow at the center of the pipeline were basically the same; that is, they were lower at the upstream

section of the capsule, while downstream of the capsule they first fluctuated greatly, then gradually decreased, and finally approached zero.

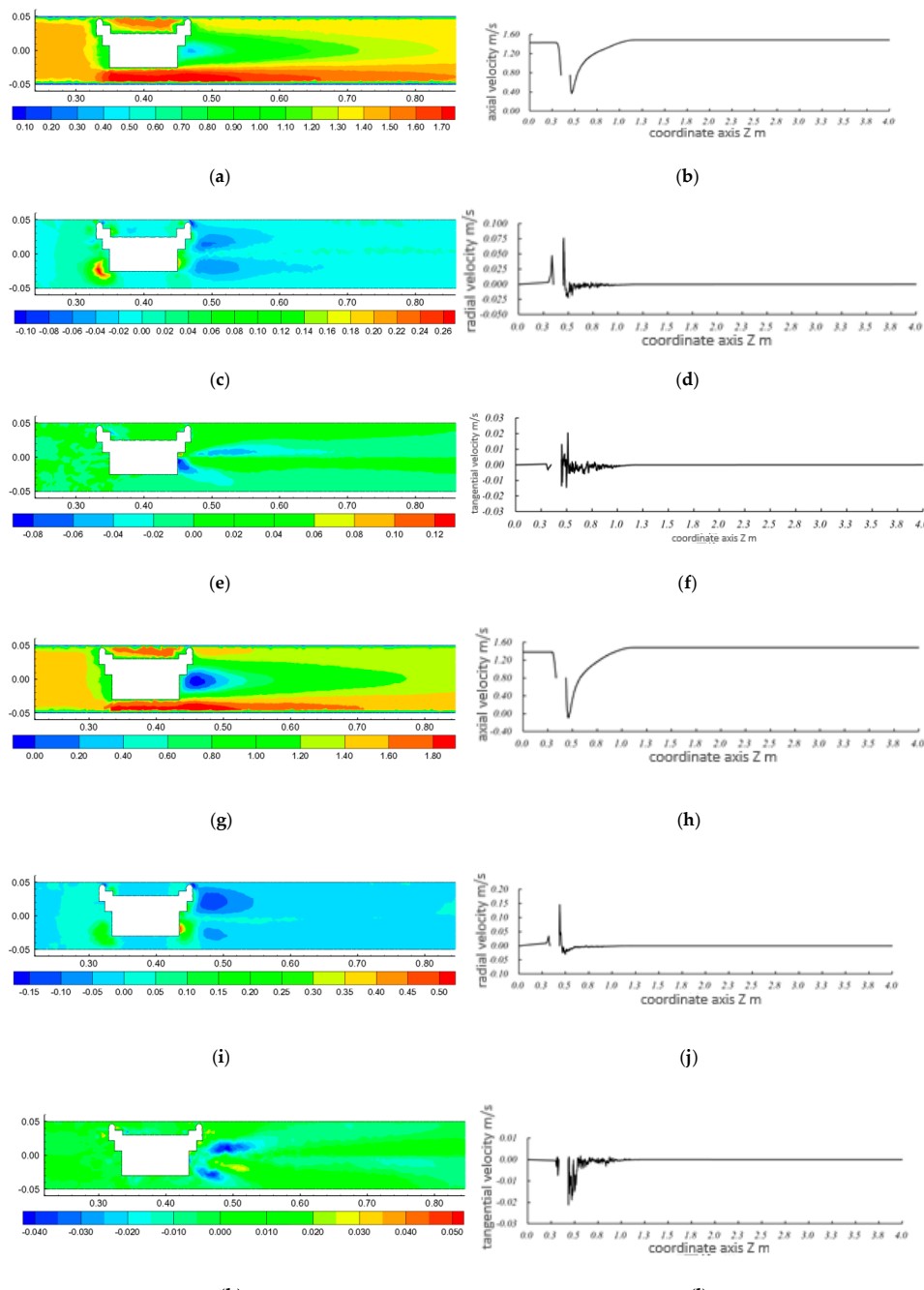

**Figure 14.** Overall flow field distribution during movement of the capsule: (**a**) axial velocity distribution in X = 0 section, *L/D* = 2; (**b**) axial velocity distribution along pipeline center points, *L/D* = 2; (**c**) radial velocity distribution in X = 0 section, *L/D* = 2; (**d**) radial velocity distribution along pipeline center points, *L/D* = 2; (**e**) circumferential velocity distribution in X = 0 section, *L/D* = 2; (**f**) circumferential velocity distribution along pipeline center points, *L/D* = 2; (**g**) axial velocity distribution in X = 0 section, *L/D* = 1.67; (**h**) axial velocity distribution along pipeline center points, *L/D* = 1.67; (**i**) radial velocity distribution in X = 0 section, *L/D* = 1.67; (**j**) radial velocity distribution along pipeline center points, *L/D* = 1.67; (**k**) circumferential velocity distribution in X = 0 section, *L/D* = 1.67; (**l**) circumferential velocity distribution along pipeline center points, *L/D* = 1.67.

## 4. Conclusions

1.  The variation trend of axial flow velocity in the downstream section of the capsule was basically the same, which was smaller in the middle area of the pipeline and larger near the inner wall. The radial flow velocity first increased and then decreased from the inner wall to the center of the pipe. The circumferential flow velocity was distributed near the support body of the capsule. At the left and right of the support body, the direction of the circumferential velocity was opposite, showing a symmetrical distribution. Moreover, the axial velocity, radial velocity, and circumferential flow velocity all increased with the increased length–diameter ratio of the capsule.

2.  The influence of the capsule on the axial velocity of the flow in the upstream section was small, only in the area near the wall of the upstream section of the capsule. The radial flow velocity first increased and then decreased from the inner wall to the center of the pipe, and the radial flow velocity in the center and the inner wall of the pipe was basically zero. The circumferential flow velocity was distributed around the support body of the capsule, showing a symmetrical distribution. Moreover, the axial velocity, radial velocity, and circumferential velocity all increased with the increased length–diameter ratio of the capsule.

3.  The axial flow velocity in the front, middle, and rear annular gap sections of the capsule first increased and then decreased from the inner wall of the pipe to the outer wall of the capsule. The axial velocity of the annular gap flow was positively correlated with the length–diameter ratio of the capsule.

4.  In the process of transporting materials, the capsule had a small influence on the upstream flow field area but a large influence on the downstream flow field area.

**Author Contributions:** Data curation, F.L. and X.Y.; Funding acquisition, X.S.; Methodology, F.L.; Resources, Y.L.; Supervision, X.S.; Writing—original draft, F.L.; Writing—review and editing, Y.L. All authors have read and agreed to the published version of the manuscript.

**Funding:** The research was funded by the National Natural Science Foundation of China (51179116, 51109155) and the Natural Science Foundation of Shanxi Province (201701D221137).

**Acknowledgments:** The research was supported by the Collaborative Innovation Center of New Technology of Water-Saving and Secure and Efficient Operation of Long-Distance Water Transfer Project at the Taiyuan University of Technology.

**Conflicts of Interest:** The authors declare that they have no conflicts of interest with respect to the research, authorship, and publication of this article.

**Data Availability:** The data used to support the findings of this study are available from the corresponding authors upon request.

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
