# Peer review of "Numerical Simulation of Flow Velocity Characteristics during Capsule Hydraulic Transportation in a Horizontal Pipe"

_water, doi:10.3390/w12041015_

Round 1
Reviewer 1 Report
This article provides the numerical results for flow velocity characteristics on capsule hydraulic transportation in a horizontal pipe. The topic is very interesting but their conclusions are prominently based on empirical results rather than the theoretical background. This referee cannot help pointing out the following comments for the completeness of the paper.
- (Major) Most ambiguous results are in subsection 3.2, which analyzes the flow velocity in the different sections of the pipe. The authors have to introduce some theoretical background for explaining the results, for example, from "Fundamentals of Two-Fluid Dynamics, D.D. Joseph, Y. Y. Renardy".
- (Major) Even if the authors make use of Fluent software, some efforts on experimental and numerical simulation for hydraulic transportation in a horizontal pipe must be introduced in the introduction section. As far as this referee knows, the followings seem to be adequate references:
"Ooms, G. et al, On the levitation force in horizontal core-annular flow with a large viscosity ratio and small density ratio, Phys. Fluids 25, 2013"
"Sotgia, G. et al, Experimental analysis of flow regimes and pressure drop reduction in oil-water mixtures, Int. J. Multiph. Flow, 2008"
"B. Lee et al, Full 3D simluations of two-phase core-annular flow in horizontal pipe using level set method, J. Sci. Comput, 2016" - (Minor) Several typos must be fixed by thorough reading.
Author Response
Dear editors and reviewers:
The serial number of this manuscript is water-750153, which is titled “Numerical simulation of flow velocity characteristics during capsule hydraulic transportation in a horizontal pipe”. First of all, we are grateful for your valuable review comments on this manuscript. Your review comments precisely pointed out several deficiencies in this manuscript, which has profound guiding significance for further modifications and improvements of this manuscript and our future research work. We will give detailed answers to the above review comments proposed by the editors and reviewers one by one below.
- Most ambiguous results are in subsection 3.2, which analyzes the flow velocity in the different sections of the pipe. The authors have to introduce some theoretical background for explaining the results, for example, from "Fundamentals of Two-Fluid Dynamics, D.D. Joseph, Y. Y. Renardy".
Answer:We have modified the part and the revision can be seen in Line 279-285 of Page 9.
- Even if the authors make use of Fluent software, some efforts on experimental and numerical simulation for hydraulic transportation in a horizontal pipe must be introduced in the introduction section. As far as this referee knows, the followings seem to be adequate references:
"Ooms, G. et al, On the levitation force in horizontal core-annular flow with a large viscosity ratio and small density ratio, Phys. Fluids 25, 2013"
"Sotgia, G. et al, Experimental analysis of flow regimes and pressure drop reduction in oil-water mixtures, Int. J. Multiph. Flow, 2008"
"B. Lee et al, Full 3D simluations of two-phase core-annular flow in horizontal pipe using level set method, J. Sci. Comput, 2016"
Answer: We have added these references and the revision can be seen in Line 88-95 of Page 2 and References[28-30].
- Several typos must be fixed by thorough reading.
Answer: I have checked the whole paper and revised them.
We have adopted red fonts to highlight the revised parts of the manuscript, which will help the editors and reviewers to review the manuscript again.
We have already answered review comments put forward by the editors and reviewers one by one in detail, and improvements and modifications have been made in the corresponding positions of this manuscript. We hope that the editors and reviewers will review the revised manuscript again. Thanks again to the editors and reviewers for their valuable review comments. If there are still any deficiencies in this manuscript, please don’t hesitate to contact me at the address below, and we will actively cooperate with the editors and reviewers to promptly modify the deficiencies in the manuscript. We deeply appreciate your consideration of our manuscript, and we are looking forward to receiving comments from the editors and reviewers.
Thank you and best regards,
Yours sincerely,
Li Fei
The first author:
Name: Li Fei
E-mail: lifei@tyut.edu.cn
Phone: +8613934239832
Address: College of Water Resource Science and Engineering, Taiyuan University of Technology, No. 79, Yingze Street, Wanbailin District, Taiyuan 030024, PR China.

Reviewer 2 Report
The paper is very interesting for readers of Water as the capsule pipelining is a promising technology in the view of emissions reduction. It is well written.
My only comment is that the references are a little bit poor, both general and technical ones. The following references are suggested:
1) at row n° 35, after the sentence:"Pipeline hydraulic transportation is a branch of pipeline transportation.", you can add: "Brandoni, C., Marchetti, B., Ciriachi, G., Polonara, F., & Leporini, M. (2016). The impact of renewable energy systems on local sustainability. International Journal of Productivity and Quality Management, 18(2-3), 385-402."
2) at row n° 41, after the sentence: "which makes it
35 one of the industries with high energy consumption and high emissions [2,3]", you can add: "Yang, X., & Ma, J. (2020). The Wall Stress of the Capsule Surface in the Straight Pipe. Water, 12(1), 242"
3) at row n° 45, after the sentence: "pollution, and no carbon emissions in the transportation process. “, you can add:” Turkowski, M., & Szudarek, M. (2019). Pipeline system for transporting consumer goods, parcels and mail in capsules. Tunnelling and Underground Space Technology, 93, 103057.”
4) at row n° 51, after the sentence: "1 during transportation and settling can occur”, you can add: “Leporini, M., Terenzi, A., Marchetti, B., Corvaro, F., & Polonara, F. (2019). On the numerical simulation of sand transport in liquid and multiphase pipelines. Journal of Petroleum Science and Engineering 175, 519-535”
Additional references are welcome.
Author Response
Dear editors and reviewers:
The serial number of this manuscript is water-750153, which is titled “Numerical simulation of flow velocity characteristics during capsule hydraulic transportation in a horizontal pipe”. First of all, we are grateful for your valuable review comments on this manuscript. Your review comments precisely pointed out several deficiencies in this manuscript, which has profound guiding significance for further modifications and improvements of this manuscript and our future research work. We will give detailed answers to the above review comments proposed by the editors and reviewers one by one below.
- at row n° 35, after the sentence:"Pipeline hydraulic transportation is a branch of pipeline transportation.", you can add: "Brandoni, C., Marchetti, B., Ciriachi, G., Polonara, F., & Leporini, M. (2016). The impact of renewable energy systems on local sustainability. International Journal of Productivity and Quality Management, 18(2-3), 385-402."
Answer: We have added this reference and the revision can be seen in Line 43 of Page1 and Reference[9].
- at row n° 41, after the sentence: "which makes it 35 one of the industries with high energy consumption and high emissions [2,3]", you can add: "Yang, X., & Ma, J. (2020). The Wall Stress of the Capsule Surface in the Straight Pipe. Water, 12(1), 242".
Answer: We have added this reference and the revision can be seen in Line 36 of Page1and Reference[4].
- at row n° 45, after the sentence: "pollution, and no carbon emissions in the transportation process. “, you can add:” Turkowski, M., & Szudarek, M. (2019). Pipeline system for transporting consumer goods, parcels and mail in capsules. Tunnelling and Underground Space Technology, 93, 103057.”
Answer: We have added this reference and the revision can be seen in Line 47 of Page2 and Reference[11].
- at row n° 51, after the sentence: "1 during transportation and settling can occur”, you can add: “Leporini, M., Terenzi, A., Marchetti, B., Corvaro, F., & Polonara, F. (2019). On the numerical simulation of sand transport in liquid and multiphase pipelines. Journal of Petroleum Science and Engineering 175, 519-535”.
Answer: We have added this reference and the revision can be seen in Line 53 of Page2 and Reference[12].
We have adopted red fonts to highlight the revised parts of the manuscript, which will help the editors and reviewers to review the manuscript again.
We have already answered review comments put forward by the editors and reviewers one by one in detail, and improvements and modifications have been made in the corresponding positions of this manuscript. We hope that the editors and reviewers will review the revised manuscript again. Thanks again to the editors and reviewers for their valuable review comments. If there are still any deficiencies in this manuscript, please don’t hesitate to contact me at the address below, and we will actively cooperate with the editors and reviewers to promptly modify the deficiencies in the manuscript. We deeply appreciate your consideration of our manuscript, and we are looking forward to receiving comments from the editors and reviewers.
Thank you and best regards,
Yours sincerely,
Li Fei
The first author:
Name: Li Fei
E-mail: lifei@tyut.edu.cn
Phone: +8613934239832
Address: College of Water Resource Science and Engineering, Taiyuan University of Technology, No. 79, Yingze Street, Wanbailin District, Taiyuan 030024, PR China.

Reviewer 3 Report
The reviewer wants to thank the authors for their paper presenting numerical investigation comparing experimental results of a capsule transported in a pipe. S/he some comments/questions/suggestions:
*1) Line (L) 81: Ulusarslan et al. [20-22] is not correct hence twice only two authors and [22] only one.
*2) L117 similar L133: Did the authors change the values of those constants? If yes, why? If no, why are that general information given, which is part of the manual/text book. At the moment it looks like the authors present this as part of the novelty and at least a reference is needed. In the reviewer’s opinion this section can be shortened and more project specific information provided.
*3) L147: The reviewer is not sure: Shouldn’t this be not just a normal dot Nm
*4) L184: The reviewer doesn’t understand the two values for the ratio. Are there two devices tested?
*5) Figure 1 and Table 1: If the authors want to use a local coordinate system, please place it at the correct origin. Wouldn’t it be easier to provide the length and diameter?
*6) Figure 1: the support body cross section is symmetrical in this sketch but obviously it can’t be hence there are only 3 connections on each side (numerical results show this correctly). Please correct this.
*7) Table 1: The reviewer assumes that all the values are in [m]. Please clarify this.
*8) Did the authors conduct a mesh study and would you please be so kind a present the results in the paper.
*9) L231: the velocity inlet is set homogeneous/constant over the complete section? L215: the “relative” pressure was set to 0 or was there a vacuum.
*10) L224: The point E is the initial condition? Please clarify this.
*11) Figure 4 and general: It is not clear how much of the experimental set-up is simulated and how the initial conditions are defined.
*12) Figure 5: Sorry, but the reviewer doesn’t understand this comparison. First how were the points measured? Where are they measured – is Zd equal to the z axis in Figure 1? Where was the capsule and was it fixed in the flow or moving?
*13) the same questions occur for the following numerical results. The reviewer is missing a key information: Is the initial rump-up phase also calculated or only a steady state? Based on the Figure 14, s/he would assume that the capsule was not moving in the flow?
The reviewer would like to clarify the mentioned points and read the paper ones more in detail. Thank you.
Author Response
Dear editors and reviewers:
The serial number of this manuscript is water-750153, which is titled “Numerical simulation of flow velocity characteristics during capsule hydraulic transportation in a horizontal pipe”. First of all, we are grateful for your valuable review comments on this manuscript. Your review comments precisely pointed out several deficiencies in this manuscript, which has profound guiding significance for further modifications and improvements of this manuscript and our future research work. We will give detailed answers to the above review comments proposed by the editors and reviewers one by one below.
- Line (L) 81: Ulusarslan et al. [20-22] is not correct hence twice only two authors and [22] only one.
Answer:We have modified it and the revision can be seen in Line 83 of Page2.
- L117 similar L133: Did the authors change the values of those constants? If yes, why? If no, why are that general information given, which is part of the manual/text book. At the moment it looks like the authors present this as part of the novelty and at least a reference is needed. In the reviewer’s opinion this section can be shortened and more project specific information provided.
Answer:We have shortened and revised the part and the revision can be seen in Line 116-137 of Page3 in Page 3. We have also added a reference and the revision can be seen in Line 116 of Page 3 and Reference[35].
- L147: The reviewer is not sure: Shouldn’t this be not just a normal dot Nm
Answer:We have modified it and the revision can be seen in Line 150 of Page4.
- L184: The reviewer doesn’t understand the two values for the ratio. Are there two devices tested?
Answer:The length–diameter ratio of the capsule refers to the ratio of the length to the diameter of the capsule. In this paper, two capsules with diameter-length ratio of 2 and 1.67 were selected for study in the same experimental device.
- Figure 1 and Table 1: If the authors want to use a local coordinate system, please place it at the correct origin. Wouldn’t it be easier to provide the length and diameter?
Answer:We have modified it and the revision can be seen in the Figure 1 and Table 1.
- Figure 1: the support body cross section is symmetrical in this sketch but obviously it can’t be hence there are only 3 connections on each side (numerical results show this correctly). Please correct this.
Answer:We have modified it and the revision can be seen in the Figure 1
- Table 1: The reviewer assumes that all the values are in [m]. Please clarify this.
Answer:The unit of all the values in Table 1 is [m] and the revision can be seen in Table 1.
- Did the authors conduct a mesh study and would you please be so kind a present the results in the paper.
Answer:We have modified the part and the revision can be seen in Line 211-217 of Page6 and Table 2.
- L231: the velocity inlet is set homogeneous/constant over the complete section? L215: the “relative” pressure was set to 0 or was there a vacuum.
Answer:The inlet velocity is the flow average velocity of the inlet section and the pressure is the relative pressure and the revision can be seen in Line 227-229 of Page7.
- L224: The point E is the initial condition? Please clarify this.
Answer:The point E is not the initial condition.
- Figure 4 and general: It is not clear how much of the experimental set-up is simulated and how the initial conditions are defined.
Answer:In this paper, numerical simulation is the main method and the experiment is mainly to verify the feasibility of the simulation method. In this paper, there are only experiments in 3.1, which are used to verify the simulation results. In 3.2, all of them are numerical simulation results. The initial conditions are defined by the velocity of the capsule.
- Figure 5: Sorry, but the reviewer doesn’t understand this comparison. First how were the points measured? Where are they measured – is Zd equal to the z axis in Figure 1? Where was the capsule and was it fixed in the flow or moving?
Answer:The velocity value of each measuring point is measured by setting the coordinates of the measuring points through the coordinate frame program. Zd represents the position of the downstream of the capsule along the z-axis. The capsule is moving in this paper.
- the same questions occur for the following numerical results. The reviewer is missing a key information: Is the initial rump-up phase also calculated or only a steady state? Based on the Figure 14, s/he would assume that the capsule was not moving in the flow?
Answer:In this paper, we simulate the flow velocity distribution in the pipe when the capsule is moving at a constant velocity. Figure 14 shows the flow velocity distribution in the pipe when the capsule is moving to a certain position in the pipe.
We have adopted red fonts to highlight the revised parts of the manuscript, which will help the editors and reviewers to review the manuscript again.
We have already answered review comments put forward by the editors and reviewers one by one in detail, and improvements and modifications have been made in the corresponding positions of this manuscript. We hope that the editors and reviewers will review the revised manuscript again. Thanks again to the editors and reviewers for their valuable review comments. If there are still any deficiencies in this manuscript, please don’t hesitate to contact me at the address below, and we will actively cooperate with the editors and reviewers to promptly modify the deficiencies in the manuscript. We deeply appreciate your consideration of our manuscript, and we are looking forward to receiving comments from the editors and reviewers.
Thank you and best regards,
Yours sincerely,
Li Fei
The first author:
Name: Li Fei
E-mail: lifei@tyut.edu.cn
Phone: +8613934239832
Address: College of Water Resource Science and Engineering, Taiyuan University of Technology, No. 79, Yingze Street, Wanbailin District, Taiyuan 030024, PR China.

Round 2
Reviewer 3 Report
The reviewer wants to thank the authors for their corrections and answers. Some points should be addressed:
*1) old point 1 and new line (L) 83: The citation is not corrected in the correct way. It should be “Ulusarslan and Teke [24,25] as well as Ulusarslan [26]”
*2) Please write in L189 the length-diameter ration of the two investigated capsules were L/D…
*3) old point *6) Figure 1. The reviewer hoped that the authors move the coordinate system into the origin. The support structure of the capsule is slightly changed but based on the reviewers understanding the top one should touch the wall in this cut and the bottom ones represents the projection of the other two. Please also correct this in the Figure 12.
*4) Please explain the numerical approach in detail. The reviewer is still not 100% sure how the authors conducted the numerical simulations. Is the fluid domain shown in Figure 1 constant and has the capsule the same distance from the inlet or is it really moving? In the first case, please clarify this and in the second case could you please describe how the relative speed of the capsule was measured and ensured that the gained results are steady state.
Thank you.
Author Response
Dear editors and reviewers:
The serial number of this manuscript is water-750153, which is titled “Numerical simulation of flow velocity characteristics during capsule hydraulic transportation in a horizontal pipe”. First of all, we are grateful for your valuable review comments on this manuscript. Your review comments precisely pointed out several deficiencies in this manuscript, which has profound guiding significance for further modifications and improvements of this manuscript and our future research work. We will give detailed answers to the above review comments proposed by the editors and reviewers one by one below.
*1) old point 1 and new line (L) 83: The citation is not corrected in the correct way. It should be “Ulusarslan and Teke [24,25] as well as Ulusarslan [26]”
Answer:We have modified it and the revision can be seen in Line 83 of Page2.
*2) Please write in L189 the length-diameter ration of the two investigated capsules were L/D…
Answer:We have modified it and the revision can be seen in Line 194 of Page5.
*3) old point *6) Figure 1. The reviewer hoped that the authors move the coordinate system into the origin. The support structure of the capsule is slightly changed but based on the reviewers understanding the top one should touch the wall in this cut and the bottom ones represents the projection of the other two. Please also correct this in the Figure 12.
Answer:We have modified it and the revision can be seen in the Figure 1and Figure 12.
*4) Please explain the numerical approach in detail. The reviewer is still not 100% sure how the authors conducted the numerical simulations. Is the fluid domain shown in Figure 1 constant and has the capsule the same distance from the inlet or is it really moving? In the first case, please clarify this and in the second case could you please describe how the relative speed of the capsule was measured and ensured that the gained results are steady state.
Answer: In the process of transporting materials, the movement of the capsule can be divided into three stages: acceleration stage, stable operation stage and deceleration stage. The flow velocity characteristics during the stable movement of the capsule in a horizontal pipe are studied in this paper.
The flow in the pipe is not constant during the stable movement of the capsule, and the distances between the capsules with each length–diameter ratio and the pipeline inlet are the same in the simulation process. In the Figure 14, the flow velocity characteristics in the pipe are compared when the capsules with different length diameter ratios move to the same position in the pipe. The velocity of the capsule is measured by infrared sensor. The infrared sensor is shown in Figure 4(b).
The revision can be seen in Line 130-133 of Page3 and139,141-142 of Page4 and Figure 4(b).
We have adopted red fonts to highlight the revised parts of the manuscript, which will help the editors and reviewers to review the manuscript again.
We have already answered review comments put forward by the editors and reviewers one by one in detail, and improvements and modifications have been made in the corresponding positions of this manuscript. We hope that the editors and reviewers will review the revised manuscript again. Thanks again to the editors and reviewers for their valuable review comments. If there are still any deficiencies in this manuscript, please don’t hesitate to contact me at the address below, and we will actively cooperate with the editors and reviewers to promptly modify the deficiencies in the manuscript. We deeply appreciate your consideration of our manuscript, and we are looking forward to receiving comments from the editors and reviewers.
Thank you and best regards,
Yours sincerely,
Li Fei
The first author:
Name: Li Fei
E-mail: lifei@tyut.edu.cn
Phone: +8613934239832
Address: College of Water Resource Science and Engineering, Taiyuan University of Technology, No. 79, Yingze Street, Wanbailin District, Taiyuan 030024, PR China.

Round 3
Reviewer 3 Report
Thank you very much for the corrections and answers. The quality of the paper was significantly increased. Looking forward to the publication.